# Acute Effects of Kawakawa (*Piper excelsum*) Intake on Postprandial Glycemic and Insulinaemic Response in a Healthy Population

**DOI:** 10.3390/nu14081638

**Published:** 2022-04-14

**Authors:** Farha Ramzan, Ramya Jayaprakash, Chris Pook, Meika Foster, Jennifer L. Miles-Chan, Richard Mithen

**Affiliations:** 1Liggins Institute, The University of Auckland, 85 Park Road, Grafton, Private Bag 92019, Auckland 1142, New Zealand; f.ramzan@auckland.ac.nz (F.R.); ramya.jayaprakash@auckland.ac.nz (R.J.); chris.pook@auckland.ac.nz (C.P.); meika@edibleresearch.co.nz (M.F.); 2Edible Research Ltd., Christchurch 7440, New Zealand; 3AuOra Ltd., Wakatū Incorporation, Nelson 7010, New Zealand; 4Human Nutrition Unit, School of Biological Sciences, The University of Auckland, Auckland 1024, New Zealand; j.miles-chan@auckland.ac.nz

**Keywords:** kawakawa, postprandial, plasma insulin, insulin sensitivity, Māori traditional medicine

## Abstract

Background: *Piper excelsum* (kawakawa) is an endemic shrub of Aotearoa, New Zealand, of cultural and medicinal importance to Māori. Its fruits and leaves are often consumed. These tissues contain several compounds that have been shown to be biologically active and which may underpin its putative health-promoting effects. The current study investigates whether kawakawa tea can modulate postprandial glucose metabolism. Methods: We report a pilot three-arm randomized crossover study to assess the bioavailability of kawakawa tea (BOKA-T) in six male participants with each arm having an acute intervention of kawakawa tea (4 g/250 mL water; 1 g/250 mL water; water) and a follow-up two-arm randomized crossover study to assess the impact of acute kawakawa tea ingestion on postprandial glucose metabolism in healthy human volunteers (TOAST) (4 g/250 mL water; and water; *n* = 30 (15 male and 15 female)). Participants consumed 250 mL of kawakawa tea or water control within each study prior to consuming a high-glycemic breakfast. Pre- and postprandial plasma glucose and insulin concentrations were measured, and the Matsuda index was calculated to measure insulin sensitivity. Results: In the BOKA-T study, lower plasma glucose (*p* < 0.01) and insulin (*p* < 0.01) concentrations at 60 min were observed after consumption of a high-dose kawakawa tea in comparison to low-dose or water. In the TOAST study, only plasma insulin (*p* = 0.01) was lower at 60 min in the high-dose kawakawa group compared to the control group. Both studies showed a trend towards higher insulin sensitivity in the high-dose kawakawa group compared to water only. Conclusions: Consuming kawakawa tea may modulate postprandial glucose metabolism. Further investigations with a longer-term intervention study are warranted.

## 1. Introduction

Kawakawa (*Piper excelsum* G. Forst; synonym *Macropiper excelsum* (G. Forst.) Miq.) is a shrub endemic to New Zealand [1]. Kawakawa is of cultural importance to Māori and has a history of use in traditional medicine to treat various health conditions, including diabetes, dermatological disorders, wounds, and genitourinary infections [2,3]. Chemical profiling of kawakawa vegetative tissues and kawakawa tea extracts has identified a complex mixture of secondary metabolites, many of which are associated with biological activity [4,5]. Foremost amongst these are lignans including (+)-excelsin and (+) diayangambin [6]; several amides including piperine, dihydropiperlonguminine, pellitorin and fagaramide [7]; the phenylpropanoids myristicin and elemicin; the flavone glycoside vitexin; and terpenoids such as α-pinene and camphene [4].

The reported biological activity of these compounds is consistent with the therapeutic effects ascribed to kawakawa by traditional Māori practitioners. For example, vitexin is reported to have the anti-diabetic potential [8], possibly by increasing the expression of glucose transporter-4 (GLUT4) [9] and by reducing the level of various pro-inflammatory cytokines, including TNF-α and interleukin-6 (IL-6) [10]. Amides, such as dihydropiperlonguminine increase glucose uptake in enterocyte-like Caco-2 cells [5]. Myristicin has been shown to have anti-inflammatory properties related to its inhibition of nitric oxide (NO) and pro-inflammatory cytokines such as IL-6 and IL-10a [11,12], and diayangambin, a lignan reported to be present in kawakawa, is also reported to exhibit immunomodulatory and anti-inflammatory properties [13].

While the biological activity of these compounds may be responsible for the reported health benefits of kawakawa, they may also have toxic effects if consumed in excess [14,15]. A recent study evaluated the safety of a commercially available kawakawa tea preparation through an oral toxicity test in laboratory rats [4]. In a 28 day study, the rats received daily doses of kawakawa tea equivalent to up to four cups for an adult human female with no adverse effects observed. The authors concluded that this dose of kawakawa is safe for human consumption. In the current work, we report the findings of two preliminary studies to explore whether consuming kawakawa tea can modulate postprandial glucose and insulin response following a high-glycemic breakfast; a three-arm, crossover, pilot study to assess the bioavailability of kawakawa tea [BOKA-T], and a two-arm, crossover, follow-up study to assess the impact of acute kawakawa tea ingestion on postprandial glucose metabolism in healthy human volunteers [TOAST].

## 2. Materials and Methods

### 2.1. Study Design and Population

The pilot study (BOKA-T) was a three-arm crossover study with six male participants. The follow-up study (TOAST) involved a two-arm crossover study with 30 participants (15 male and 15 female), powered on the change in postprandial glucose flux observed in BOKA-T. All participants had a BMI of 18–25 kg/m^2^. Candidates with any current or past endocrine disorders, cardiovascular diseases (CVD), cancer, pre-existing type 2 diabetes mellitus (T2DM), or taking current medications (e.g., anti-inflammatory drugs), dietary or herbal supplements that might interfere with the study outcomes were excluded. Participants were required to abstain for 24 h from any kind of dietary/edible spices and fast overnight (10–12 h) before attending the clinic.

The studies were approved by the Central Health and Disability Ethics Committee (20/CEN/69) and the Southern Health and Disability Ethics Committee (20/STH/236) for BOKA-T and TOAST, respectively. Both studies were conducted according to the Declaration of Helsinki guidelines. All participants provided written informed consent. The studies are prospectively registered with the Australian New Zealand Clinical Trials Registry at www.anzctr.org.au (accessed on 14 March 2022) (BOKA-T: ACTRN12620000629932p; and TOAST: ACTRN12621000311853).

### 2.2. Intervention Dose and Breakfast

The safe dose of kawakawa tea for human consumption was determined based on a previous study by Butts et al. [4]. For BOKA-T, two doses of kawakawa tea were prepared: a higher dose containing 4 g of kawakawa dried leaves (commercially available at ŌKU Ltd., New Zealand) infused in 250 mL of hot water (75–80 °C) and a lower dose containing 1 g of kawakawa dried leaves infused in 250 mL of hot water (75–80 °C) (Figure 1A). For TOAST, only one dose of dried 4 g kawakawa leaves (commercially available at ŌKU Ltd., Hamilton, New Zealand) infused in 250 mL of hot water (75–80 °C) was used (Figure 1B). The washout period for both studies was designed to prevent the possible carry-over effect of kawakawa metabolites from the previous study arm [16]. Since no previous human data were available on the bioavailability of kawakawa metabolites for the BOKA-T, the three interventions were administered 1 week apart. Within TOAST, the washout period was reduced to 48 h based on the observation from BOKA-T that the kawakawa metabolites were excreted within 24 h (unpublished data). The tea was strained for the BOKA-T and TOAST studies before the participants were served. All the participants were asked to consume the tea within 10 min of serving.

For both studies, breakfasts provided to the participants had a combined estimated glycemic index (GI) of >70. The GI values were sourced from a database maintained at the University of Sydney [17] and were considered to be high glycemic [18]. For BOKA-T, a 250 mL of rice milk (578 kJ) with a 30 g serving of instant oats (489 kJ) was provided. For TOAST, two slices of white bread (935 kJ), 10 g of strawberry jam (80 kJ) and 250 mL of rice milk (578 kJ) were provided.

### 2.3. Sample Collection

Fasting and postprandial venous blood were collected in EDTA-coated vacutainer tubes from each participant and immediately stored on ice before processing. For BOKA-T, postprandial blood samples were collected across the 24 h period at 30, 45, 60, 90, 120, 180, 240, and 300 min, and 24 h. For TOAST, postprandial blood samples were collected over 3 h at 30, 45, 60, 90, 120, and 180 min. Plasma was separated and aliquoted within 2 h of collection by centrifugation at 1900× *g* for 15 min at 4 °C and was immediately stored at −80 °C until further analysis.

### 2.4. Anthropometric and Biochemical Analysis

Height (Seca 222, Hamburg, Germany) and bodyweight (Mettler Toledo Spider, Zurich, Switzerland) were recorded with lightly clad participants without shoes. Waist circumference was measured midway between the inferior margin of the lower rib and the iliac crest using a non-stretchable metal tape. On arrival, participants were asked to rest for a few minutes before their blood pressure (BP; mmHg) was recorded using a digital Critikon Dinamap Sphygmomanometer (GE Healthcare, Shanghai, China). Biochemical measures of baseline and postprandial plasma glucose were obtained by enzymatic colorimetric assay (Roche Diagnostics Ltd., Mannheim, Germany) using a Hitachi 902 autoanalyser (Hitachi High Technologies Corporation, Japan). Plasma insulin was measured by the microparticle enzyme immunoassay (Cobas Elecsys^®^, 2010; Roche Diagnostics, Basel, Switzerland) using Cobas Modular P800 (Roche Diagnostics, Auckland, New Zealand). Plasma Lipids including total cholesterol, low-density lipoprotein cholesterol (LDL-C), high-density lipoprotein cholesterol (HDL-C) and triglycerides were measured using Cobas Modular P800 (Roche Diagnostics, Auckland, New Zealand).

### 2.5. Insulin Sensitivity Assessment

Using fasting and postprandial plasma glucose and insulin measurements, the whole-body insulin sensitivity index (ISI) was calculated using the equation of Matsuda and De-Fronzo [19]:1000/√Glucose_0_ × Insulin_0_ × Glucose_mean_ × Insulin_mean._

### 2.6. Metabolomic Analysis

Metabolic profiling of kawakawa tea was performed using liquid chromatography with tandem mass spectrometry [LC–MS/MS]. Briefly, kawakawa dried leaves were infused in hot water (80 °C) for ten minutes. After extraction, the samples were rapidly cooled on ice to room temperature, followed by centrifugation to remove any particulate materials. The liquid supernatant was then analysed for the metabolite profiles of the kawakawa.

### 2.7. Statistical Analysis

As no previous human intervention studies were available on the effects of kawakawa consumption on blood glucose and insulin concentrations, the sample size for BOKA-T was estimated according to the phase 1 ICH-GCP clinical trial guidelines of having a minimum of 6 subjects [20]. For the TOAST study, the required sample size was estimated as the number of participants required to detect a “physiologically meaningful” decrease in the postprandial glucose change at the 60 min (the primary variable) of 5%, a type I error (α) of 0.05 and the desired power (1-β) of 0.90 as observed in BOKA-T. Using these data, a sample size of 26 participants per group was estimated; however, we increased recruitment to 30 per group to maintain statistical power in the event of participant drop-outs; 15 males and 15 females were therefore recruited.

All of the data are shown as the mean ± SEM. Samples with a value of more than three times the interquartile range were treated as outliers and were subsequently removed from further analysis [21]. Participants with fasting glucose > 5.6 mmol/L were excluded from the analysis [22]. For BOKA-T, a two-way, mixed-models, repeated-measures ANOVA was conducted to understand the effect of treatment and time on plasma glucose, insulin and lipid concentrations. For TOAST, a three-way, mixed-models, repeated-measures ANOVA was used to understand the effects of sex, treatment and time on plasma glucose, insulin and lipid concentrations. Statistical significance of the main effect was accepted at a Bonferroni-adjusted alpha level of 0.025. Pairwise comparisons were performed for statistically significant main effects. Non-parametric paired signed-rank test (Wilcoxon test) was used to measure the differences in the ISI. The incremental AUC (iAUC) of plasma glucose, insulin and lipid concentration was calculated using the trapezoid method, including positive and negative peaks, using GraphPad Prism-8 (GraphPad Software, San Diego, CA, USA). All analyses were carried out using SPSS 25.0 (SPSS Inc., Chicago, IL, USA), and all graphs were prepared using GraphPad Prism-8.

## 3. Results

### 3.1. Participant Characteristics

Participant clinical and demographic characteristics for both the BOKA-T and TOAST studies are summarised in Table 1. All six participants (male) completed the BOKA-T study, while as in the TOAST study, one participant (male) discontinued after the first visit, with only 29 of the 30 (14 male and 15 female) completing the study. The data of the discontinued participant were excluded from all data analyses. On analysing the study data, three further participants (male) in the TOAST study had elevated fasting glucose values (>5.6 mmol/L) and therefore were excluded from any data analyses (i.e., the sample size for analysis was 26 (11 male and 15 female).

### 3.2. Metabolomic Profiling of Kawakawa Tea

Identities to the features of collected metabolomic data were assigned based on the retention times and mass spectra obtained in LC–MS/MS scans. Most of the features in the data matched metabolites previously reported in the literature describing kawakawa phytochemistry. The most abundant features in the data were identified to belong to four different classes of metabolites, including phenylpropanoids (e.g., myristicin and elemicin), lignans (yangambin), and alkyl amides (fagaramide). Detailed results of the metabolomics profiling will be reported separately but were largely similar to those reported by Butts et al. [4].

### 3.3. Postprandial Plasma Glucose Response in the BOKA-T and TOAST Studies

Plasma glucose concentrations as a function of time during the postprandial period are presented in Figure 2 for each of the BOKA-T and TOAST studies, respectively. All values were within expected concentrations for healthy individuals. No differences in fasting plasma glucose between different intervention groups and sex were observed in either of the two studies. For BOKA-T, plasma glucose concentrations showed significant group and time differences (main effect = *p* < 0.01 both factors), with a lower glucose level in the 4 g/250 mL kawakawa tea group at 60 min following the meal. In TOAST, the plasma glucose concentration showed significant differences for both groups (*p* = 0.04) and time (*p* < 0.01) as main effects; however, no effect of treatment on glucose concentrations at any time point was observed. A significant main effect of sex was observed on the postprandial glucose concentrations (*p* = 0.01), with no significant effect of treatment or timepoint was observed. The iAUC for glucose did not differ significantly between the two groups.

### 3.4. Postprandial Plasma Insulin Response in the BOKA-T and TOAST Studies

Plasma insulin concentrations as a function of time during the postprandial period are presented in Figure 3. No differences in fasting plasma insulin between different intervention groups and sex were observed in either of the two studies. With regard to BOKA-T, a significant effect of time (*p* < 0.01) was observed; both the groups who consumed tea (1 g/250 mL and 4 g/250 mL) observed a lower increase in insulin concentrations at 60 min post-intervention (post-tea) compared to hot water only. In TOAST, a significant main effect of time was observed (*p* < 0.01) and, like BOKA-T, a significant two-way interaction was observed between time and treatment group (*p* = 0.01). Individuals consuming 4 g/250 mL kawakawa tea showed a lower increase in the postprandial insulin concentrations at 60 min compared to the control. A significant main effect of sex was observed on the postprandial insulin concentrations (*p* = 0.02), with no significant interaction with treatment or timepoint was observed. The iAUC for insulin did not differ significantly between the two groups.

### 3.5. Postprandial Plasma Lipids Response in the BOKA-T and TOAST Studies

In BOKA-T samples, a significant effect of treatment (*p* < 0.01) was observed between the groups for all analysed blood lipid parameters, including HDL-C, LDL-C, total cholesterol and triglycerides (Figure 4A).

In TOAST, the main effect of both the treatment and time was observed in total cholesterol (*p* < 0.01 and *p* = 0.02, respectively) and HDL-C (*p* < 0.01 and *p* = 0.01) between the two groups; however, only a main effect of treatment was observed in plasma LDL-C (*p* < 0.01) and triglycerides (*p* < 0.01). Individuals consuming 4 g/250 mL kawakawa tea showed a lower increase in the postprandial triglyceride concentrations than the controls (Figure 4B). In addition, a significant interaction between sex and treatment was observed (*p* = 0.01), with females consuming 4 g/250 mL kawakawa tea having a lower increase in postprandial triglyceride concentrations than male. The iAUC did not differ significantly between conditions for any of the postprandial lipid parameters, except that iAUC for triglycerides was close to a level of statistical significance (*p* = 0.08).

### 3.6. Insulin Sensitivity Index

While the Matsuda index is considered a good indicator of insulin sensitivity [19], there is no agreed threshold value for detecting insulin resistance, although an upper threshold of four has been suggested [23,24,25]. Based on this threshold, in the BOKA-T study, we observed an increase in ISI in four participants consuming kawakawa tea compared to control (Figure 5A). In TOAST, compared to the control intervention, 19 of the 25 participants observed an increase in the ISI following kawakawa tea consumption (Figure 5B). As such, no significant differences were observed in the median ISI values of the two groups in BOKA-T using the Wilcoxon test; however, in TOAST, a significant increase in the ISI was observed following consumption of 4 g/250 mL kawakawa tea compared to the water control (*p* = 0.05; median difference = 0.46).

## 4. Discussion

Despite the traditionally ascribed benefits of kawakawa consumption on health, no reported human studies on health-related biomarkers exist. In both the BOKA-T and TOAST studies, individuals who consumed tea made with 4 g of kawakawa dried leaves before a high-glycemic breakfast had reduced postprandial plasma insulin concentrations and increased insulin sensitivity compared to the water control. Within the BOKA-T study, we also observed a decrease in the postprandial plasma glucose and a dose-related kawakawa response in postprandial insulin, with tea made using 4 g kawakawa significantly decreasing postprandial insulin levels compared to the lower dose of 1 g kawakawa, or water only. Additionally, in the TOAST study, a lower increase in postprandial plasma triglycerides was observed in participants consuming 4 g/250 mL kawakawa tea prior to a high-glycemic meal compared to water control

Postprandial glucose and insulin dysregulation are independent risk factors for obesity and related cardio-metabolic diseases [26,27]. Although the mechanisms underpinning the decreased postprandial insulin levels with kawakawa intake are not known, there is a probability that kawakawa intake might affect the first stage of insulin-stimulated glucose uptake. Various bioactives in kawakawa, such as isovitexin, vitexin, and piperine, have affected glucose homeostasis. For example, isovitexin and vitexin present in mistletoe fig (*Ficus deltoidea*) have been reported to affect glucose metabolism by enhancing GLUT 4 receptor activity [9]. A recent in vitro study observed that piperine-related compounds from kawakawa fruit increased glucose uptake and decreased fatty acid uptake in differentiated intestinal Caco-2 cells [7]. Additionally, kawakawa has a complex chemistry, and there is likely an additive or synergistic effect between these compounds.

Considering the anti-hyperglycemic effect of several phytochemicals in kawakawa, its intake could also affect insulin release from the β-cells in the pancreas. For example, a study by Folador et al. [28] involving oral administration of taiuiá (*Wilbrandia ebracteata* Cogn.) in diabetic and non-diabetic rats observed potential secretagogue effects of isovitexin in non-diabetic rats, suggesting that isovitexin and related flavonoids demonstrate anti-hyperglycemic effects by regulating the insulin secretion or related functions in the pancreas [28]. Vitexin, on the other hand, is shown to improve insulin signalling in non-alcoholic fatty liver disease (NAFLD) mice through upregulation of insulin receptor substrate-1 (IRS-1) and also its downstream target AKT [29].

It has been demonstrated that insulin sensitivity and secretion are mutually related, with an improved insulin sensitivity resulting in reduced β-cell sensitivity and decreased peripheral insulin levels [30]. Previous studies have reported improved insulin sensitivity and reduced postprandial insulin response following consumption of several dietary components [31,32]. For instance, green tea extracts affect glycemic control by improving peripheral insulin sensitivity. In a study by Venables et al. [33], a 13% increase in the Matsuda measured insulin sensitivity was observed in response to acute ingestion of green tea extract in healthy males compared to the placebo. Although the exact mechanism for the observed effects of green tea is not yet known, it is hypothesised that green tea acts via glucose transporters in tissues such as skeletal muscle, thus requiring less insulin to clear a given amount of glucose [34]. Similarly, in the TOAST study, an 8.3% higher insulin sensitivity in the 4 g/250 mL kawakawa group compared to control, demonstrating a possible role of kawakawa in regulating pathways related to insulin sensitivity. However, determining whether kawakawa has a potential effect on the mechanisms involved in glycemic control that could lead to reduced insulin and improved insulin sensitivity is beyond this paper’s scope and would require further mechanistic studies.

Prediabetic and diabetic patients often present an exaggerated postprandial triglyceride response after meals [35,36]. This undesirable postprandial response is related to the diminished insulin sensitivity of different body tissues [36]. In the TOAST study, lower postprandial plasma triglycerides were observed in participants who consumed 4 g/250 mL kawakawa tea prior to a high-glycemic meal than control. This observed response to kawakawa intake could be attributed to various alkamides and piperine analogues in kawakawa. For example, pellitorin is shown to have an anti-adipogenic effect by activating the transient receptor potential vanilloid type 1 (TRPV1) [37]. Piperine, and related compounds, impact intestinal fatty acid and glucose absorption [5]. Identifying the compounds in kawakawa tea that exert the biological effects observed here, such as the decrease in plasma triglycerides, remains a subject for further research. However, this is an interesting observation, suggesting that kawakawa intake might be involved in regulating both postprandial insulinaemic and lipidaemic responses.

Several limitations indicate the need for careful interpretation of the study results. Firstly, individuals with overweight or obesity display dysregulated markers of blood glycemic response; however, both the BOKA-T and TOAST studies only included healthy participants with a BMI of 18–25 kg/m^2^. Thus, it is difficult to evaluate from our studies whether the observed effects on the glycemic response by kawakawa intake may translate into the same effects in the metabolically dysregulated individuals. Secondly, these preliminary studies involved only the acute exposure to kawakawa and did not investigate the mechanism of action of kawakawa. Therefore, whether the observed acute glycemic changes by kawakawa tea may translate into long-term anti-diabetic effects remains unknown and warrants longer and larger studies that directly address the impact of kawakawa involving a metabolically dysregulated population. Nevertheless, whatever the mechanisms underlying the observed effects exerted by kawakawa tea on postprandial insulin concentrations and sensitivity, our data were consistent in both the pilot and follow-up study and suggests a potential benefit derived from kawakawa intake on postprandial insulin response.

## 5. Conclusions

In summary, acute intake of kawakawa tea (4 g/250 mL hot water) attenuated insulin response to a subsequent high-glycemic meal. Although the BOKA-T study observed a lower increase in postprandial plasma glucose with the high dose of kawakawa intake compared to the control, no similar effects were observed in the TOAST study. Our preliminary findings suggest that kawakawa consumption may improve insulin response and sensitivity. However, further investigations with longer-term interventions are required to assess any physiological or clinical significance.

## Figures and Tables

**Figure 1 nutrients-14-01638-f001:**
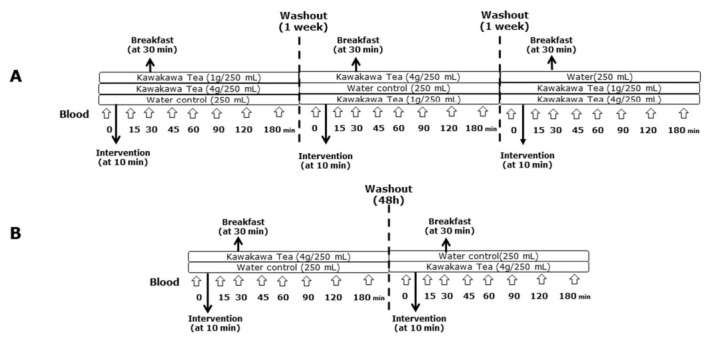
Overview of the study design and intervention. (**A**) A three-arm crossover study (BOKA-T; *n* = 6) involving three groups; 0 g (only hot water), 1 g (1 g kawakawa/250 mL hot water) and 4 g (4 g kawakawa/250 mL hot water). (**B**) A two-arm crossover study (TOAST; *n* = 30) involving two groups; 0 g (only hot water) and 4 g (4 g kawakawa/250 mL hot water).

**Figure 2 nutrients-14-01638-f002:**
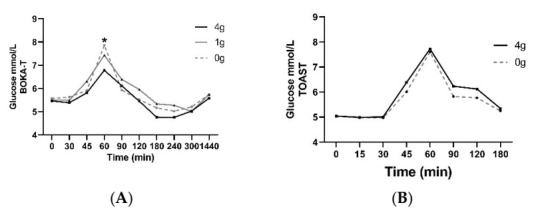
Mean ± SEM plasma concentrations of glucose for BOKA-T (**A**) and TOAST (**B**) throughout the postprandial period; 0 g (only hot water), 1 g (1 g kawakawa/250 mL hot water) and 4 g (4 g kawakawa/250 mL hot water). Differences between the treatment group over the postprandial were determined using repeated-measures ANOVA (* indicates *p* ≤ 0.05).

**Figure 3 nutrients-14-01638-f003:**
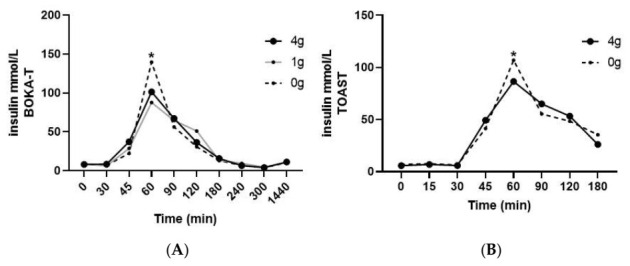
Mean ± SEM plasma concentrations of insulin for BOKA-T (**A**), and TOAST (**B**) throughout the postprandial period; 0 g (only hot water), 1 g (1 g kawakawa/250 mL hot water) and 4 g (4 g kawakawa/250 mL hot water). Differences between the treatment group over the postprandial were determined using repeated-measures ANOVA (* indicates *p* ≤ 0.05).

**Figure 4 nutrients-14-01638-f004:**
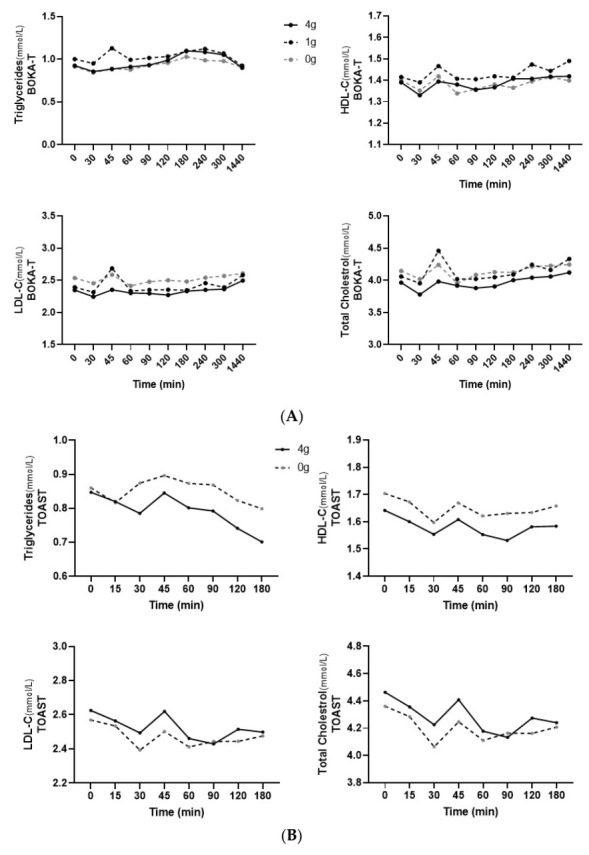
(**A**) Mean ± SEM plasma concentrations of plasma lipids for BOKA-T throughout the postprandial period. (**B**) Mean ± SEM plasma concentrations of plasma lipids for TOAST throughout the postprandial period; 0 g (only hot water), 1 g (1 g kawakawa/250 mL hot water) and 4 g (4 g kawakawa/250 mL hot water). Differences between the treatment group over the postprandial were determined using repeated-measures ANOVA.

**Figure 5 nutrients-14-01638-f005:**
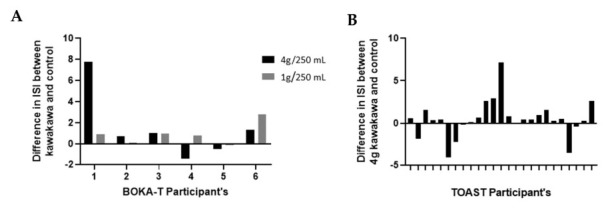
The difference in the insulin sensitivity index (Matsuda index) of kawakawa in comparison to control group: (**A**) BOKA_T: 4 g/250 mL kawakawa and 1 g/250 mL kawakawa in comparison to control (250 mL hot water); (**B**) TOAST: 4 g/250 mL kawakawa in comparison to control (250 mL hot water).

**Table 1 nutrients-14-01638-t001:** Baseline clinical and demographic characteristics of participants for the BOKA-T and TOAST studies.

Variables	BOKA-T	TOAST		
	Male	Female	Male	All
Total number (n)	6	15	14	29
Age (year)	31.2 ± 2.7	30.9 ± 1.3	32.0 ± 1.6	32.0 ± 1.7
Body weight (kg)	72.0 ± 2.7	58.8 ± 1.8	68.0 ± 1.8	68.2 ± 1.7
BMI (kg/m^2^)	23.8 ± 0.8	22.3 ± 0.3	22.3 ± 0.4	22.4 ± 0.3
Systolic BP (mmHg)	120 ± 4.5	109 ± 2.1	117 ± 3.1	114 ± 2.9
Diastolic BP (mmHg)	77 ± 4.2	73 ± 1.9	75 ± 2.0	75 ± 2.1

Data are shown as the mean ± SE; BMI: body mass index; BP: blood pressure.

## Data Availability

The data presented in this study are available on request from the corresponding author. The data are not publicly available due to ethical concerns.

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
