# Peer review of "Acute Effects of Kawakawa (*Piper excelsum*) Intake on Postprandial Glycemic and Insulinaemic Response in a Healthy Population"

_nutrients, 2022, doi:10.3390/nu14081638_

Round 1

Reviewer 1 Report

The manuscript provides novel observations on the effects of kawakawa tea on the postprandial responses for glucose and insulin. Experimental studies on the effects of kawakawa tea in humans are limited. I have only a few comments for clarification purposes.

I suggest to define BOKA-T and TOAST in the abstract.

Ls 88-92. Why is there a range of water temperatures? Will the hot water temperature having an effect on bioavailability, see also L142.

L204. “A significant sex effect was observed on the postprandial glucose concentrations (p=0.01).”Please elaborate on the comparison between males and females.

L222. “A significant sex effect was observed on 222 the postprandial insulin concentrations (p=0.02). Please elaborate on the comparison between males and females.

The legend of Figure 4A and B has “(* indicates p≤0.05).” but no * are in the figures. Please revise.

L258. Replace “has been” with “has been suggested”.

Figure 5. provide figures with an A and B.

Author Response

  1. I suggest to define BOKA-T and TOAST in the abstract.

We thank the reviewer for these suggestions, we have now updated the manuscript to define the acronyms BOKA-T and TOAST in the abstract

  1. Ls 88-92. Why is there a range of water temperatures? Will the hot water temperature having an effect on bioavailability, see also L142.

We thank the reviewer for commenting on the water temperature range for brewing tea. We agree that a wide range of temperatures could influence the bioavailability of the tea metabolites. However, in ideal situations maintaining the exact 80 degrees in the clinical unit was not possible as compared to lab situations. We managed to maintain the temperature range between 75-80 degrees which was incorrectly typed in the manuscript as 70-80. It is possible that there may be minor changes to the bioactive compounds due to this variation, but with unlikely physiological consequences

  1. “A significant sex effect was observed on the postprandial glucose concentrations (p=0.01).”Please elaborate on the comparison between males and females.

We thank the reviewer for this comment. We only observed a main effect of sex on the postprandial glucose levels, with no significant interaction with the treatment or timepoint. We have accordingly updated the text in the manuscript "line 206-208".

  1. “A significant sex effect was observed on 222 the postprandial insulin concentrations (p=0.02). Please elaborate on the comparison between males and females.

We thank the reviewer for this comment. As with glucose, we only observed a main effect of sex on the postprandial insulin levels, with no significant interaction with the treatment or timepoint. We have accordingly updated the text in the manuscript "line 225-227".

  1. The legend of Figure 4A and B has “(* indicates p≤0.05).” but no * are in the figures. Please revise.

We thank the reviewer for the careful reading of the text, we have accordingly updated the figure legend.

  1. Replace “has been” with “has been suggested”.

We thank the reviewer for the comment, we have updated the text.

  1. Figure 5. provide figures with an A and B..

We thank the reviewer for the careful reading of the text, we have accordingly updated the figure numbers.

Reviewer 2 Report

This paper presents an important study where consuming kawakawa tea was shown to give higher insulin sensitivity compared to water only.  For more convincing discussion it would have been interesting to see the kawakawa tea extract compositions (1g and 4g/250ml).  The introduction gives kawakawa compositions but it does not include the tea compositions as for the solubility of bioactive compounds might be different in tea. 

Author Response

We thank the reviewer for their careful reading of the manuscript and for these thoughtful comments. As stated in the manuscript, we undertook a metabolite analyses of the kawakawa tea (paragraphs 2.6 and 3.2) and found the composition of both the 1g /250ml and 4g/250ml tea similar to that previously reported by Butts et al 2019.  We are currently undertaking a more detailed chemical analysis which will be reported separately.